# Haloperoxidase-Catalyzed Luminol Luminescence

**DOI:** 10.3390/antiox11030518

**Published:** 2022-03-08

**Authors:** Robert C. Allen

**Affiliations:** Department of Pathology, Creighton University, Omaha, NE 68178, USA; robertallen@creighton.edu; Tel.: +1-402-350-3193

**Keywords:** haloperoxidase, myeloperoxidase, eosinophil peroxidase, horseradish peroxidase, halide oxidation, singlet molecular oxygen, luminol luminescence, chemiluminescence, reaction order, kinetic analysis

## Abstract

Common peroxidase action and haloperoxidase action are quantifiable as light emission from dioxygenation of luminol (5-amino-2,3-dihydrophthalazine-1,4-dione). The velocity of enzyme action is dependent on the concentration of reactants. Thus, the reaction order of each participant reactant in luminol luminescence was determined. Horseradish peroxidase (HRP)-catalyzed luminol luminescence is first order for hydrogen peroxide (H_2_O_2_), but myeloperoxidase (MPO) and eosinophil peroxidase (EPO) are second order for H_2_O_2_. For MPO, reaction is first order for chloride (Cl^−^) or bromide (Br^−^). For EPO, reaction is first order for Br^−^. HRP action has no halide requirement. For MPO and EPO, reaction is first order for luminol, but for HRP, reaction is greater than first order for luminol. Haloperoxidase-catalyzed luminol luminescence requires acidity, but HRP action requires alkalinity. Unlike the radical mechanism of common peroxidase, haloperoxidases (XPO) catalyze non-radical oxidation of halide to hypohalite. That reaction is second order for H_2_O_2_ is consistent with the non-enzymatic reaction of hypohalite with a second H_2_O_2_ to produce singlet molecular oxygen (^1^O_2_*) for luminol dioxygenation. Alternatively, luminol dehydrogenation by hypohalite followed by reaction with H_2_O_2_ yields dioxygenation consistent with the same reaction order. Haloperoxidase action, Cl^−^, and Br^−^ are specifically quantifiable as luminol luminescence in an acidic milieu.

## 1. Introduction

Myeloperoxidase (MPO) enzymatic action produces light emission or chemiluminescence. Such luminescence is native, i.e., no chemiluminigenic substrate is needed, requires H_2_O_2_, halide, and acidic pH [1,2], and correlates with the requirements for MPO microbe killing described by Klebanoff [3]. Both luminescence and microbicidal action are hydrogen peroxide (H_2_O_2_), chloride (Cl^−^), and acid-dependent. Light emission implies haloperoxidase-catalyzed combustive oxygenation. MPO catalyzes H_2_O_2_ oxidization of halide to hypohalite, e.g., Cl^−^ oxidation to hypochlorite (OCl^−^). The non-enzymatic reaction of hypohalite with a second H_2_O_2_ produces electronically excited singlet molecular oxygen (^1^O_2_*) [4,5,6]. Both H_2_O_2_ and OCl^−^ are singlet multiplicity reactants necessitating a single multiplicity product [7,8]. The relaxation of ^1^O_2_* to its triplet ground state (^3^O_2_) requires intersystem crossing, and as such, ^1^O_2_* has about a microsecond lifetime [9,10]. This lifetime is sufficient for reactivity, but such reaction is restricted to within a radius of about 0.3 microns (micrometer) of its nascence.

Spin conservation and frontier orbital considerations restrict the direct reaction of ground-state triplet multiplicity oxygen (^3^O_2_) with singlet multiplicity biomolecules. Thus, combustion is not spontaneous, and the large exergonicity that would result from such action is unrealized. Spin and frontier orbital considerations do not limit ^1^O_2_* reaction with biomolecules. The electrophilic reactivity of ^1^O_2_* drives oxygenation of organic molecules, and a fraction of the reaction products will have electronically excited singlet multiplicity carbonyl functions that relax by emitting photons in the visible spectrum.

The originally described chemiluminescence measurements of MPO activity did not involve use of a chemiluminigenic probe. However, application of luminol (5-amino-2,3-dihydrophthalazine-1,4-dione) as a substrate increases the sensitivity for detecting cellular oxygenation activities by about a thousand-fold [11]. Luminol is an established chemiluminigenic probe for the study of common peroxidase activity [12,13,14] and has also been applied to the study of haloperoxidase action [15]. The common peroxidase action of horseradish peroxidase (HRP) has been extensively studied, and the research findings support a radical reaction pathway [16].

Luminol dioxygenation creates electronically excited singlet aminophthalate (3-aminophthalate*) that relaxes to ground state by photon emission [17]. Whether by peroxidase or haloperoxidase action, luminol dioxygenation must satisfy spin conservation and frontier orbital requirements. This report documents the unique character of haloperoxidase action with respect to reaction order. Reaction requires contact, and the probability of contact increases with the concentration of the reactants. For a given reactant, the rate or velocity of reaction is dependent on its order of reaction [18]. The following presents and contrasts the orders of the reactants responsible for haloperoxidase and peroxidase catalyzed luminol luminescence.

## 2. Materials and Methods

Myeloperoxidase (MPO) and eosinophil peroxidase (EPO) were purified from porcine leukocytes by Exoxemis, Inc. MPO concentration was assessed based on an absorbance extinction coefficient (ε) at 430 nm (ε_430_ nm) of 178 mM^−1^∙cm^−1^ [19]. The rheinheitzahl (RZ) using a 430 to 280 nm absorbance ratio (A_430/280_) was 0.7. EPO was measured using an absorbance extinction coefficient at 412 nm (ε_412_ nm) of 110 mM^−1^∙cm^−1^ [20]. The RZ (A_412/280_) for EPO was 0.7. Horseradish peroxidase (HRP) was purchased from Sigma Chemical Co. and measured by direct absorbance using an absorbance extinction coefficient of 90 mM^−1^∙cm^−1^ at 403 nm [21]. Hydrogen peroxide, luminol, and other chemicals were purchased from Sigma-Aldrich Chemical. The concentration of luminol was determined using an absorbance extinction coefficient (ε_351_ nm) of 7.6 mM^−1^∙cm^−1^ [22]. Appreciate that the solubility of luminol as well as its quantum yield decrease with increasing acidity [22,23].

Spectrophotometric measurements were on a DW2000 UV-visible spectrophotometer (SLM Instruments Co., Urbana, IL, USA) or on an Agilent 8453 UV-visible diode array spectrophotometer (Agilent Technologies, Santa Clara, CA, USA). Light was measured using an AutoLumat LB 953 Multi-tube Luminometer equipped with two reagent injectors (Berthold Technologies GmbH & Co., Bad Wildbad 75323, Germany). Measurements were at ambient temperature 22 ± 2 °C. The luminometer reports light emission in relative light units (RLU). Luminescence velocity or intensity measurements are described as kilocounts per second (kcts/s = kiloRLU/s) or as kilocounts per 10 s (kcts/10 s) as indicated. Reactions were in 50 mM acetate buffer adjusted to the pH indicated or 50 mM phosphate buffer adjusted to the pH indicated. Reaction was initiated by mixing enzyme, halide, and luminol solutions with H_2_O_2_ solution in a polystyrene tube at a final volume of 0.9 or 1.0 mL as indicated. The final concentrations of reactants and enzymes are expressed in molar concentration terms described as millimolar (millimole/L or mM), micromolar (µmole/L or µM), nanomolar (nanomole/L or nM), picomolar (picomole/L or pM), or for halides, milliequivalent/L (mEq/L). Data storage and analysis used MS Excel and/or SSPS software.

## 3. Results

### 3.1. Reaction Order with Respect to H_2_O_2_

Luminol luminescence requires dioxygenation. For haloperoxidase action, the first step in the oxygenation process is the H_2_O_2_-dependent oxidation of halide to hypohalite, and the second step is the non-enzymatic oxidation of H_2_O_2_ by hypohalite generating ^1^O_2_*. The third step of the process is reaction of ^1^O_2_* with luminol resulting in dioxygenation and light emission. These collective reactions are consistent with spin conservation and frontier orbital reaction rules. Both H_2_O_2_ and hypohalite are singlet multiplicity reactants, as is the product ^1^O_2_*. Although only one H_2_O_2_ is required for haloperoxidase-catalyzed halide oxidation, a total of two H_2_O_2_ molecules are required to produce ^1^O_2_*. An alternative pathway that is also spin conserved could involve direct hypohalite dehydrogenation of luminol followed by reaction with an additional H_2_O_2_ resulting in dioxygenation and luminescence [24]. Such reaction also requires two H_2_O_2_ molecules, i.e., reaction remains second order for H_2_O_2_.

Reaction rate depends on the concentration of each reactive participant. Figure 1 depicts the correlation of luminol luminescence velocity with H_2_O_2_ concentration [H_2_O_2_]. The ordinate (*y*-axis) describes luminescence velocity in kilocounts/s, and the abscissa (*x*-axis) describes the mM concentration of H_2_O_2_. The haloperoxidase is 78 nM MPO, the halide is 100 mEq/L Cl^−^ and the pH is 5.0. Note that luminol luminescence velocity increases with H_2_O_2_ concentration up to about 6 mM before decreasing. H_2_O_2_ inhibits MPO at relatively high concentrations [25,26].

The data of Table 1 and the composite plots of Figure 2 illustrate that MPO-catalyzed luminol luminescence is second order with respect to [H_2_O_2_]. Figure 2A shows luminescence velocity in the 0 to 6.3 mM range of [H_2_O_2_]. Instead of a hyperbolic curve that would be typical for a first-order reactant, the curve shows sigmoidal character. Measuring the velocity at low concentrations of H_2_O_2_ allows direct assessment of reaction order. In the less than 1 mM H_2_O_2_ concentration range, the luminescence velocity equals the H_2_O_2_ concentration taken to the exponential 1.95, i.e., velocity = 335 × [H_2_O_2_]^1.95^ with a coefficient of determination (R^2^) of 0.991. MPO-catalyzed luminol luminescence is second order with respect to [H_2_O_2_], i.e., [H_2_O_2_]^2^. Figure 2B depicts the typical double reciprocal plot of the data where the *y*-axis is expressed as 1/velocity and *x*-axis is expressed as 1/[H_2_O_2_] concentration. If velocity was first order with respect to substrate, double reciprocal plotting of 1/velocity against 1/substrate would linearize the hyperbolic geometry and produces a straight line. This is clearly not the case for H_2_O_2_. The rate equation is second order with respect to H_2_O_2_, and accordingly, the double reciprocal plot is not linearized. 

Figure 2C illustrates that the double reciprocal plot of 1/velocity against the reciprocal of the [H_2_O_2_] squared (1/[H_2_O_2_]^2^) does yield a linear relationship. Figure 2D shows that plotting the *y*-axis as the reciprocal square root of the velocity (1/√velocity) against 1/[H_2_O_2_] also yields a linear relationship. Haloperoxidase-catalyzed luminol luminescence is second order with respect to H_2_O_2,_ revealing its distinct difference from common peroxidase action that is first order with respect to H_2_O_2_.

### 3.2. Reaction Order with Respect to Cl^−^ or Br^−^

MPO catalyzes the H_2_O_2_-dependent oxidization of Cl^−^ and Br^−^ producing hypochlorus acid (HOCl) and hypobromus acid (HOBr), respectively. Figure 3 reveals the characteristic hyperbolic curves observed with Cl^−^ (Figure 3A) or Br^−^ (Figure 3C) as halide. At low halide concentrations, the luminol luminescence velocity approximates a linear relationship with regard to halide concentration. In the range up to 5 mEq/L Cl^−^, the luminescence velocity in kcts/s equals 157.5 × [Cl^−^]^0.86^ with an R^2^ of 0.999. Consistent with this first-order relationship, MPO-catalyzed oxidation of either Cl^−^ or Br^−^ shows linearity by double reciprocal plotting, i.e., 1/v = (K_m_/V_max_) × 1/sub + 1/V_max_, where 1/v is the reciprocal luminescence velocity (1/velocity), the slope is the apparent Michaelis constant (K_m_) divided by the maximum velocity (V_max_), 1/sub is the reciprocal substrate concentration (1/[Cl^−^]) and the y-intercept is 1/V_max_ [18]. The double reciprocal plots for Cl^−^ and Br^−^ are presented as Figure 3B,D, respectively. Haloperoxidase-catalyzed oxidation of halide is first order with respect to either Cl^−^ or Br^−^. The MPO-catalyzed reaction is first order for halide and first order for H_2_O_2_, but as will be discussed subsequently, the composite reactions required for luminol luminescence are second order for H_2_O_2_.

Although the data are not presented, high halide:H_2_O_2_ ratios exert a competitive inhibitory action on haloperoxidase-catalyzed luminol luminescence. The presence of halide also functions to protect haloperoxidase from damage [27].

### 3.3. EPO Catalyzes Br^−^ but Not Cl^−^ Oxidation

EPO catalyzes the H_2_O_2_-dependent oxidization of Br^−^ but does not catalyze oxidation of Cl^−^ measured as luminol luminescence. Figure 4A shows that Cl^−^ activity is minuscule compared to the Br^−^ activity shown in Figure 4C. The Br^−^ data of Figure 4C show suitable linearity (R^2^ = 0.998) on double reciprocal plotting in Figure 4D. Double reciprocal plotting of the Cl^−^ data presented in Figure 4B does not show linearity (R^2^ = 0.187). The reagent-grade NaCl used to prepare the Cl^−^ concentrations has trace Br^−^, typically less than 0.01% to 0.001% Br^−^, that could contribute to the slight increase in luminescence with increasing concentrations of Cl^−^.

### 3.4. Reaction Order with Respect to Luminol

Dioxygenation of luminol produces electronically excited singlet multiplicity 3-aminophthalate* that relaxes to ground state by photon emission. Various pathways are available for such dioxygenation, and examination of the reaction order provides insight as to the pathway taken. The orders of reaction with respect to luminol were investigated for MPO, EPO, and HRP. MPO plus 100 mEq/L Cl^−^ and EPO plus 10 mEq/L Br^−^ were assayed at pH 5. HRP showed essentially no activity at pH 5, but HRP activity was measurable at pH 7 without halide.

The plots on the left side of Figure 5 show luminescence velocities against luminol concentration for MPO (Figure 5A), EPO (Figure 5C), and HRP (Figure 5E) in the upper, middle, and lower frames of the composite graphic, respectively. At concentrations below 10 µM luminol, the luminescence velocity is approximately linear with respect to luminol concentration. The data were collected from two similar luminol concentration experiments. The luminescence velocity equations regarding luminol concentration were calculated to be y = 58 × [luminol]^0.911^ (R^2^ = 0.996) and y = 48 × [luminol]^0.982^ (R^2^ = 0.997) for MPO, and y = 21 × [luminol]^0.780^ (R^2^ = 1.0) and y = 19 × [luminol]^0.696^ (R^2^ = 1.0) for EPO. MPO and EPO catalyzed luminescence approximate first order for luminol. The rate equations for HRP luminescence velocity using luminol concentrations less than 10 µM were y = 18 × [luminol]^1.128^ (R^2^ = 0.997) and y = 17 × [luminol]^1.197^ (R^2^ = 1.0). Note that HRP catalyzed luminescence is greater than first order for luminol.

The right side of Figure 5 presents the double reciprocal plots of the data for MPO (Figure 5B), EPO (Figure 5D), and HRP (Figure 5F) as the upper, middle, and lower frames, respectively. The MPO and EPO catalyzed activities are approximately first order with respect to luminol concentration, and consequently, the double reciprocal plots expressed as 1/velocity for the ordinate and as 1/luminol for the abscissa show the expected linearity (Figure 5B,D). However, best linear fit for HRP catalyzed activity for luminol is realized when the double reciprocal plot expresses the ordinate as 1/√velocity and the abscissa as 1/luminol (Figure 5F). This result is consistent with the findings reported by Dure and Cormier [13].

### 3.5. Acid Versus Alkaline Activities for MPO, EPO, and HRP

The haloperoxidase actions of MPO and EPO directed to microbe killing and detoxification are optimal in a mild acid milieu, and such action can be quantified by luminol luminescence. Appreciate that MPO and EPO also show luminol luminescence by common peroxidase action at alkaline pH.

Luminol luminescence measurements of the haloperoxidase and the common peroxidase actions of MPO, EPO, and HRP were quantified at pH 5 and pH 8.2 in the presence of Cl^−^ or Br^−^. The left upper graphic of Figure 6 shows luminol luminescence velocity plotted against MPO concentration for 100 mEq/L Cl^−^ and for 2.5 mEq/L Br^−^ at a pH of 5.0 (Figure 6A), and the right upper graphic (Figure 6B) show MPO measured under the same conditions except at a pH of 8.2. Alkalinity increases the total luminescence and diminishes the difference between Br^−^ versus Cl^−^. Common peroxidase action is not halide dependent.

The left middle graphic of Figure 6 shows luminol luminescence velocity plotted against EPO concentration for 100 mEq/L Cl^−^ and for 2.5 mEq/L Br^−^ at a pH of 5.0 (Figure 6C). At pH 5.0, the haloperoxidase action of EPO catalyzes H_2_O_2_ oxidation of Br^−^ but does not catalyze H_2_O_2_ oxidization of Cl^−^. Consequently, Cl^−^ show essentially no luminescence response. The right middle graphic shows EPO activities under the same conditions except for a pH of 8.2 (Figure 6D). Alkalinity favors luminol luminescence by common peroxidase action and also increases the chemiluminescent quantum yield of luminol [22]. Common peroxidase action does not require Br^−^ or Cl^−^. Note that luminescence is now observed with Cl^−^ but does not require Cl^−^.

The left bottom graphic of Figure 6 shows luminescence velocity plotted against HRP concentration for 100 mEq/L Cl^−^ and for 2.5 mEq/L Br^−^ at a pH 5.0 (Figure 6E). HRP is incapable of haloperoxidase action and shows no luminol luminescence for either halide at pH 5.0. As shown in the right bottom graphic (Figure 6F), HRP shows proper common peroxidase-catalyzed luminol luminescence at pH 8.2, and this action is halide independent.

## 4. Discussion

Common peroxidase action involves radical one-equivalent transfer reactions, ultimately resulting in the dehydrogenation of substrate, or in the case of luminol, dioxygenation [28,29]. Regarding luminol luminescence, such peroxidase action is first order with respect to H_2_O_2_, greater than first order with respect to luminol, has no halide requirement, and requires an alkaline milieu. Haloperoxidase-catalyzed luminol luminescence requires the two-equivalent reduction in H_2_O_2_ and the two-equivalent oxidation of Cl^−^ or Br^−^ to OCl^−^ or OBr^−^ with the haloperoxidase returned to its original redox state [1,30,31]. The initial haloperoxidase-catalyzed reaction is expected to be first order for H_2_O_2_ and first order for halide, but the composite reactions responsible for luminol luminescence are expected to be and are second order for H_2_O_2_, first order for halide, and first order for luminol.

The results presented in Figure 6 demonstrate that haloperoxidase-catalyzed luminol luminescence requires an acid milieu, whereas common peroxidase catalyzed luminol luminescence requires an alkaline milieu. This pH differs from that requirement for peroxidase-catalyzed dye oxidation. Common peroxidase action measured as dye dehydrogenation occurs over a broad pH range. Peroxidase-catalyzed guaiacol dehydrogenation is optimum at mildly acid pH, but proper peroxidase action is measured over a range of pH from 4 to 8 [32]. Regarding optimal pH, peroxidase action measured by dye dehydrogenation differs from peroxidase action measured by luminol luminescence.

The results presented employed luminol as a high quantum yield substitute for the various oxidizable biomolecules that comprise microbes. Luminol oxygenation produces luminescence by both common peroxidase action and haloperoxidase action, but the pathways to dioxygenation differ, as demonstrated by the fundamental distinctions in the orders for the participant reactants. Mild acidity, e.g., pH 5, selects for haloperoxidase luminol luminescence action, and such luminol luminescence is second order for H_2_O_2_, first order for halide, and first order for luminol. Haloperoxidase-catalyzed luminol luminescence requires combining the haloperoxidase reaction to a non-enzymatic reaction, with each reaction consuming a molecule of H_2_O_2_. The overall reaction is second order for H_2_O_2_ and is consistent with the generation of ^1^O_2_*. The second order for H_2_O_2_ result can also be explained by hypohalite dehydrogenation of luminol followed by reaction with an additional H_2_O_2_ [24]. However, the second order for H_2_O_2_ result is not consistent with any mechanism of common peroxidase action.

Neutrophil leukocytes are the phagocyte effectors of microbicidal action. Phagocytosis activates neutrophil NADPH oxidase [33], facilitating the reduction in triplet multiplicity oxygen (^3^O_2_) and ultimately supplying the two H_2_O_2_ molecules that drive MPO action and the generation of ^1^O_2_* [34]. Its triplet ground state restricts the reaction of ^3^O_2_ with the singlet multiplicity molecules of biochemistry [8]. The actions of NADPH oxidase and MPO effectively change the spin quantum number of oxygen, allowing frontier orbital reactivity and oxygenation of the singlet multiplicity molecules that embody biology. Although ^1^O_2_* is a potent electrophilic reactant, its reactivity is constrained by its microsecond lifetime that limits combustive action to the vicinity of MPO-bound microbe [35].

## 5. Conclusions

Analysis of MPO and EPO-catalyzed luminol luminescence shows that reaction is:Second order with respect to H_2_O_2_, i.e., y = k × [H_2_O_2_]^2^ where y is luminescence velocity and [H_2_O_2_]^2^ is the square of the H_2_O_2_ concentration,

∴ to realize linearity, double reciprocal enzymatic plot of the velocity must be in the form 1/square root of velocity (1/√velocity) for the ordinate and 1/[H_2_O_2_] for the abscissa.

First order with respect to Cl^−^ for MPO, i.e., y = k × [Cl^−^]^1^ or y = mx + b, where y is luminescence velocity and [x] is Cl^−^ concentration, [Cl^−^]^1^;First order with respect to Br^−^ for MPO and EPO, i.e., y = k × [ Br^−^]^1^ or y = mx + b, where y is luminescence velocity and [x] is Br^−^ concentration, [Br^−^]^1^;First order with respect to luminol for MPO and EPO, i.e., y = k × [luminol]^1^ or y = mx + b, where y is luminescence velocity and [x] is luminol concentration, [luminol]^1^;First order or linear with respect to MPO and EPO concentration.

Haloperoxidase activity measured as luminol luminescence requires an acidic milieu. EPO is unable to catalyze Cl^−^ oxidation.

The research presented also confirms previous reports on HRP catalyzed luminol luminescence. Common peroxidase activity is:First order with respect to H_2_O_2_, i.e., y = k × [H_2_O_2_]^1^ or y = mx + b, where y is luminescence velocity and [x] is the H_2_O_2_ concentration.Greater than first order with respect to luminol, i.e., y = k × [luminol]^1+^, where y is luminescence velocity and [luminol]^1+^ is luminol concentration,

∴ to realize linearity, double reciprocal enzymatic plot of the velocity is best achieved by plotting 1/square root of velocity (1/√velocity) for the ordinate and 1/[luminol] for the abscissa.

First order or linear with respect to HRP concentration.

HRP does not catalyze Cl^−^ or Br^−^ oxidation. Although peroxidase-catalyzed dye dehydrogenation occurs over a broad acid to alkaline pH range, optimal peroxidase-catalyzed luminol luminescence action requires an alkaline milieu.

Like the native MPO luminescence associated with microbicidal action, the haloperoxidase-catalyzed luminol luminescence of MPO requires mild acidity, is halide dependent, and can be considered in three steps [1,2]:
H_2_O_2_ + Cl^−^ ― (MPO) → H_2_O + OCl^−^Step 1H_2_O_2_ + OCl^−^ ―(non-enzymatic) → H_2_O + Cl^−^ + ^1^O_2_*Step 2^1^O_2_* + microbe (luminol) → combustive dioxygenation → luminescenceStep 3

Microbes present various molecular compositions with differing susceptibilities to electrophilic oxygenation, but some fraction of these combustive dioxygenations yield products with electronically excited singlet carbonyl functions that relax by photon emission as observed in native luminescence. Luminol serves as the high quantum yield molecular substitute for the heterogeneous molecules of the microbe, but even luminol has limited solubility and quantum efficiency at an acidic pH of about 5 [22,23].

Step 1 describes MPO haloperoxidase action driving H_2_O_2_ oxidation of Cl^−^ producing OCl^−^ and is first order for H_2_O_2_ and first order for Cl^−^. Step 2 describes the non-enzymatic oxidation of H_2_O_2_ by OCl^−^ generating ^1^O_2_* [4] and is also first order for H_2_O_2_ and first order for OCl^−^. This Step 2 reaction has a rate constant of 4.4 × 10^7^ M^−1^∙s^−1^ [5]. Cl^−^ behaves like an enzyme cofactor and is consumed in Step 1 and replenished in Step 2. Low concentration of H_2_O_2_ favors chloramine formation. However, chloramines also react with H_2_O_2_ to generate ^1^O_2_* [36] with rate constants in the range of 10^4^ to 10^6^ M^−1^∙s^−1^ [6]. The temporary capture of hypochlorite as chloramine, followed by chloramine reaction with H_2_O_2_ to produce ^1^O_2_*, has the same orders of reaction for H_2_O_2,_ Cl^−^, and luminol. Step 3 describes the reaction of ^1^O_2_* with luminol yielding dioxygenation and luminescence. In combination, Steps 1, 2, and 3 generate ^1^O_2_*, dioxygenation and luminescence. Such haloperoxidase-catalyzed luminol luminescence is second order for H_2_O_2_, first order for Cl^−^ and first order for luminol. An alternative spin-allowed, non-radical pathway with the same orders of reaction can result from hypochlorite dehydrogenation of luminol followed by reaction of the dehydrogenated intermediate with H_2_O_2_ yielding dioxygenation and photon emission [37,38].

The results presented demonstrate that haloperoxidase action can be specifically quantified as luminol luminescence in an appropriate acidic milieu and that the presence of MPO and EPO can be quantified and differentiated based on controlling Cl^−^ and Br^−^ in such measurement. Likewise, the presence of Cl^−^ and Br^−^ can be quantified and differentiated based on the halide dependence of MPO and EPO-catalyzed luminol luminescence in an acidic milieu.

## Figures and Tables

**Figure 1 antioxidants-11-00518-f001:**
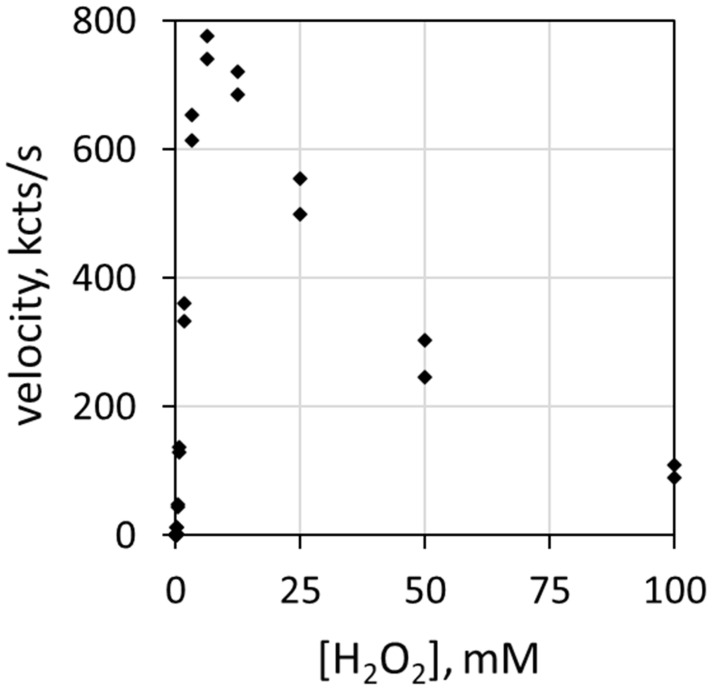
Plot of luminescence velocity against H_2_O_2_ concentration using 78 nM MPO with 100 mEq/L Cl^−^ and 51 µM luminol in 50 mM acetate buffer at pH 5.0. Reaction was initiated by contact mixing of all components. The final volume was 0.9 mL.

**Figure 2 antioxidants-11-00518-f002:**
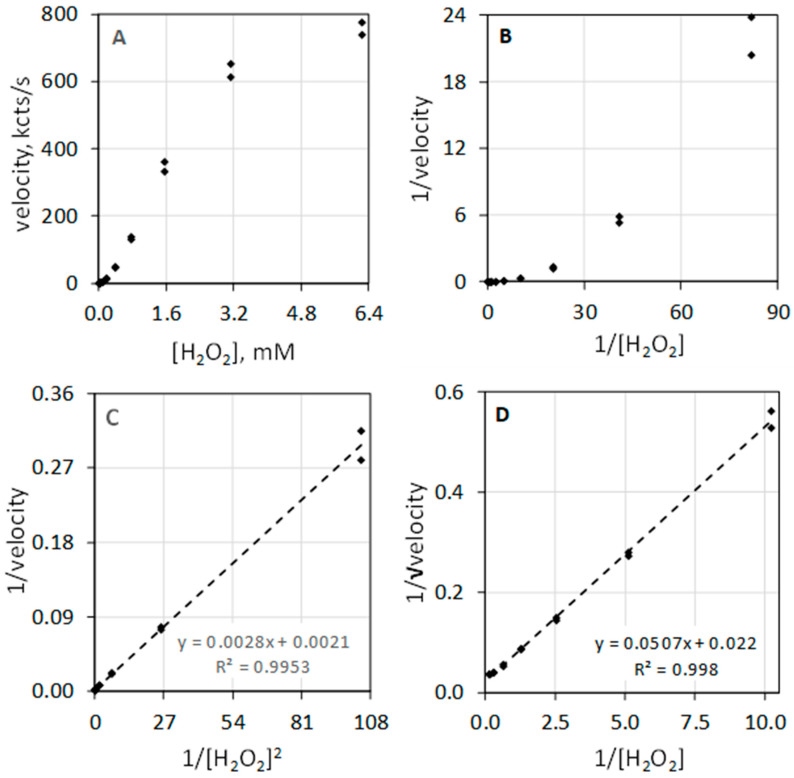
Luminescence velocity plotted against H_2_O_2_ under conditions as described in Figure 1. (**A**) depicts the plot of velocity against H_2_O_2_ to a concentration of 6.3 mM. (**B**) shows the standard double reciprocal (Lineweaver–Burk) plot of 1/velocity (1/(kcts/s)) against 1/[H_2_O_2_] (1/mM). (**C**) shows that linearity is achieved when the reciprocal of velocity (1/(kcts/s)) is plotted against the 1/[H_2_O_2_]^2^ (1/(mM)^2^). (**D**) shows that linearity is also achieved when the reciprocal of the square root of the velocity (1/√velocity) (1/√(kcts/s)) is plotted against 1/[H_2_O_2_] (1/mM).

**Figure 3 antioxidants-11-00518-f003:**
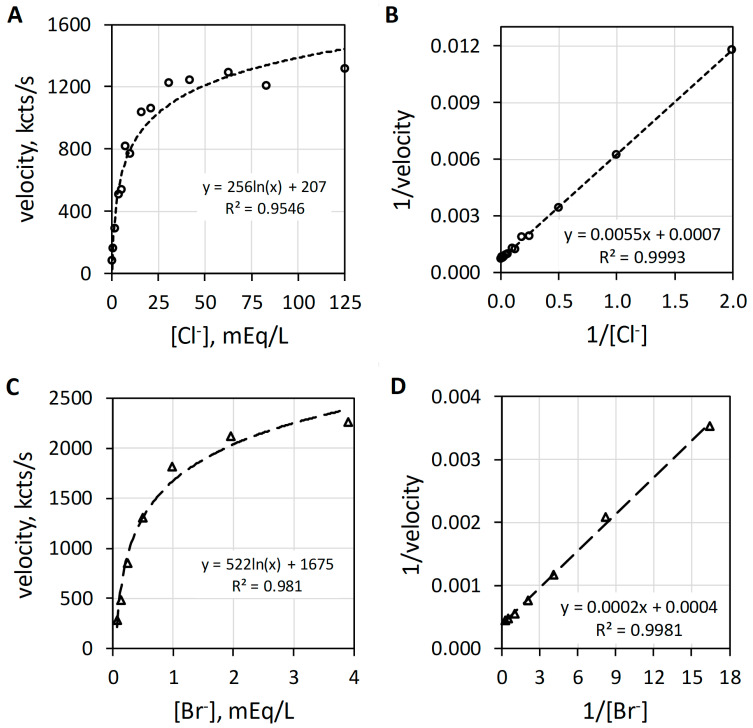
Plots of MPO-catalyzed luminol luminescence against halide concentration expressed in milliequivalents per liter (mEq/L). (**A**) shows the hyperbolic relationship of velocity to increasing [Cl^−^] (mEq/L). (**B**) presents the double reciprocal plot of 1/velocity (1/(kcts/s)) against 1/[Cl^−^] (1/(mEq/L)) showing proper linearity. (**C**) depicts the hyperbolic response to increasing [Br^−^] (mEq/L). (**D**) shows the double reciprocal plot of 1/velocity (1/(kcts/s)) against 1/[Br^−^] (1/(mEq/L)) showing suitable linearity. Cl^−^ and Br^−^ concentrations were varied with a constant concentration of 78 nM MPO, 51 µM luminol, and 6.3 mM H_2_O_2_ in 50 mM acetate buffer pH 5. Reaction was initiated by mixing of reactants. The final volume was 1.0 mL.

**Figure 4 antioxidants-11-00518-f004:**
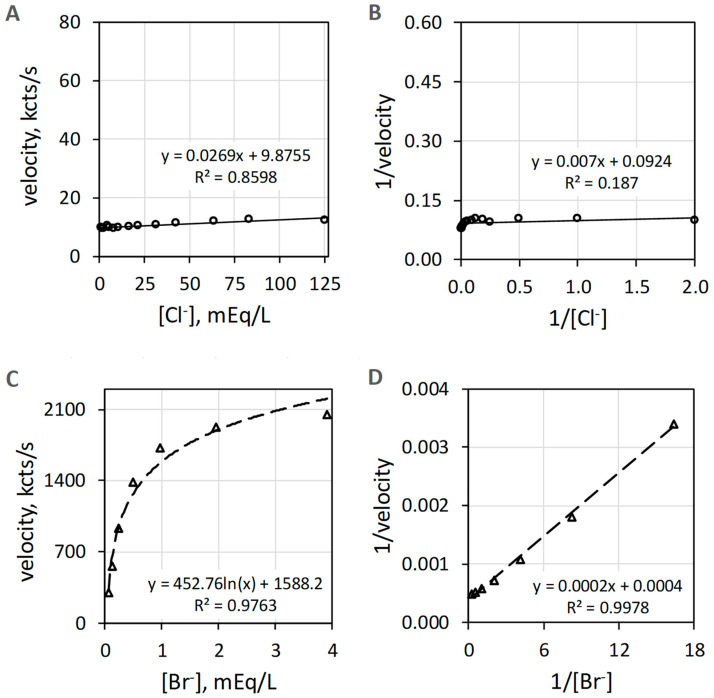
Plots of EPO catalyzed luminol luminescence against halide concentration. (**A**) shows very low velocity to all [Cl^−^] (mEq/L) tested. (**B**) shows the double reciprocal plot of 1/velocity (1/(kcts/s)) against 1/[Cl^−^] (1/(mEq/L)) with a coefficient of determination (R^2^) of 0.19. (**C**) depicts the hyperbolic response to increasing [Br^−^] (mEq/L). (**D**) shows the double reciprocal plot of 1/velocity (1/(kcts/s)) against 1/[Br^−^] (1/(mEq/L)) showing proper linearity. Cl^−^ and Br^−^ concentrations were varied using a constant concentration of 39 nM EPO, 51 µM luminol, and 6.3 mM H_2_O_2_ in 50 mM acetate buffer pH 5. Reaction was initiated by mixing of reactants. The final volume was 1.0 mL. Background luminescence in the absence of enzyme is about 1.5 kcts/s.

**Figure 5 antioxidants-11-00518-f005:**
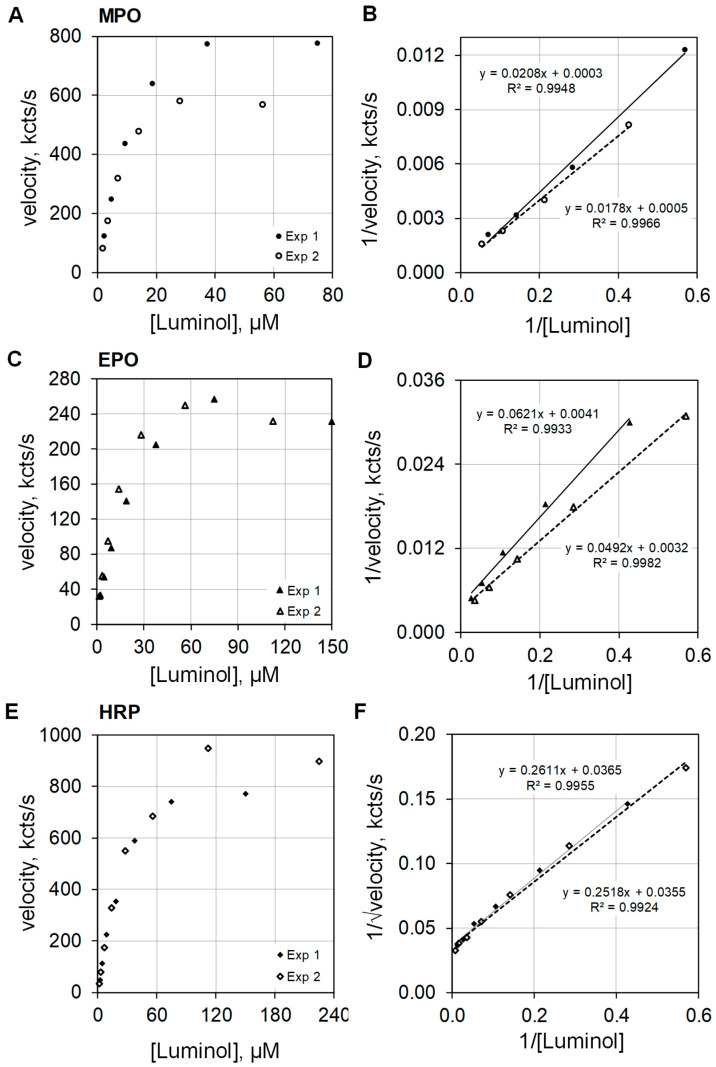
Plots of luminescence velocity against luminol concentration for MPO, EPO, and HRP. The top graphics present the results using 100 nM MPO with 100 mEq/L Cl^−^ (**A**), the middle graphics present the results using 100 nM EPO with 10 mEq/L Br^−^ (**C**), and the bottom graphics present the results using 100 nM HRP without halide (**E**). The plots to the right show double reciprocal plots of 1/velocity (1/(kcts/s)) against 1/[luminol] (1/(µM)) for MPO (**B**) and EPO (**D**). For HRP, the double reciprocal plots show 1/√velocity (1/√(kcts/s)) against 1/[luminol] (1/(µM)) (**F**). MPO and EPO activities were measured in 50 mM acetate buffer at pH 5. HRP activity was measured in 50 mM phosphate buffer at pH 7. Reaction was initiated by mixing reactants with 2.5 mM H_2_O_2_. The final volume was 1.0 mL.

**Figure 6 antioxidants-11-00518-f006:**
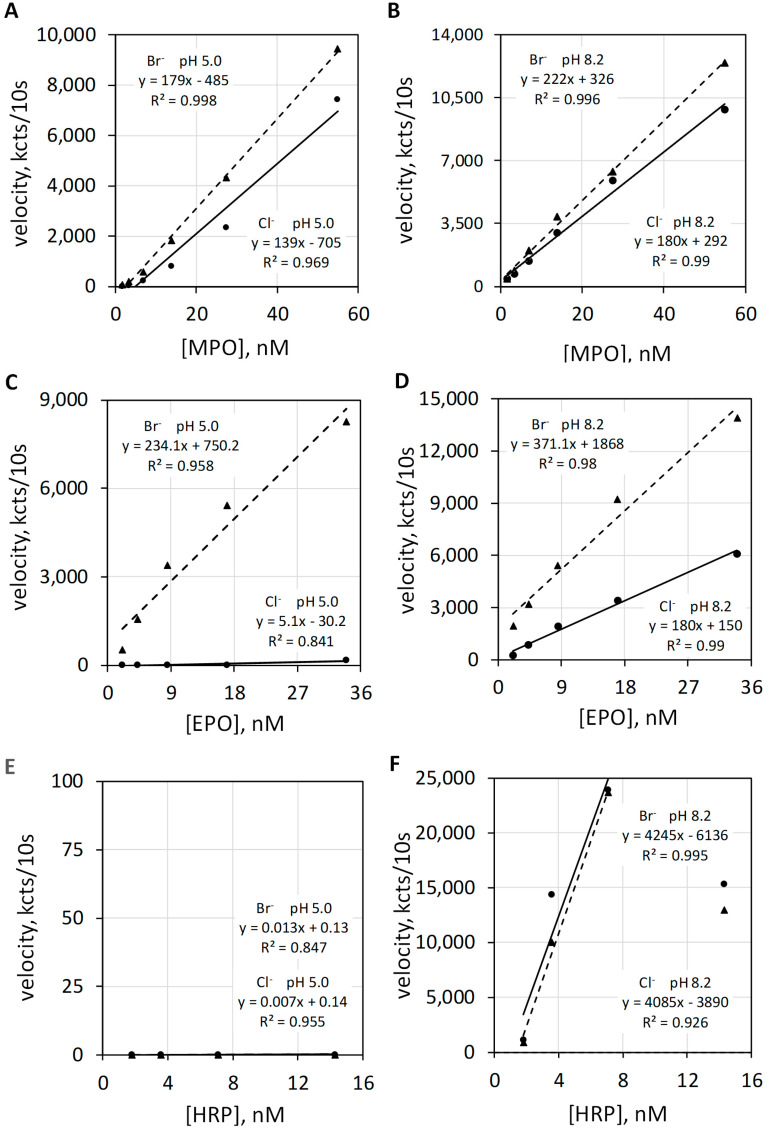
Luminol luminescence velocity plotted against MPO, EPO, and HRP concentration under acid and alkaline conditions. Varying concentrations of MPO (**A**,**B**), EPO, (**C**,**D**) and HRP (**E**,**F**) activities were measured in 50 mM acetate buffer at pH 5.0 (**A**,**C**,**E**) and in 50 mM phosphate buffer at pH 8.2 (**B**,**D**,**F**). MPO, EPO, and HRP were measured with 100 mEq/L Cl^−^ and with 2.5 mEq/L Br^−^. The luminol concentration was 50 µM. Reaction was initiated by mixing reactants with 2.5 mM H_2_O_2_. The final volume was 1.0 mL. Velocity was measured as kilocounts collected during the initial 10 s post mixing and expressed as kcts/10 s.

**Table 1 antioxidants-11-00518-t001:** Myeloperoxidase-catalyzed luminol luminescence dependence on H_2_O_2_ concentration.

H_2_O_2_	[H_2_O_2_]^2^	Velocity (v)	√v	1/[H_2_O_2_]	1/[H_2_O_2_]^2^	1/v	1/√v
mM	mM^2^	kcts/s	√(kcts/s)	1/mM	1/mM^2^	1/kcts/s	1/√(kcts/s)
6.250	39.062	739.83	27.20	0.16	0.026	0.001	0.037
3.126	9.769	614.07	24.78	0.32	0.102	0.002	0.040
1.563	2.444	333.59	18.26	0.64	0.409	0.003	0.055
0.781	0.610	130.08	11.41	1.28	1.639	0.008	0.088
0.391	0.153	44.82	6.69	2.56	6.537	0.022	0.149
0.196	0.038	12.78	3.57	5.11	26.149	0.078	0.280
0.098	0.010	3.18	1.78	10.23	104.597	0.315	0.561
0.049	0.002	0.76	0.87	20.45	418.388	1.309	1.144
0.024	0.001	0.17	0.41	40.91	1673.554	5.848	2.418
0.012	0.000	0.04	0.20	81.82	6694.215	23.810	4.880
6.250	39.062	775.88	27.85	0.16	0.026	0.001	0.036
3.126	9.769	653.23	25.56	0.32	0.102	0.002	0.039
1.563	2.444	361.17	19.00	0.64	0.409	0.003	0.053
0.781	0.610	137.72	11.74	1.28	1.639	0.007	0.085
0.391	0.153	47.91	6.92	2.56	6.537	0.021	0.144
0.196	0.038	13.46	3.67	5.11	26.149	0.074	0.273
0.098	0.010	3.58	1.89	10.23	104.597	0.279	0.528
0.049	0.002	0.82	0.91	20.45	418.388	1.214	1.102
0.024	0.001	0.19	0.43	40.91	1673.554	5.319	2.306
0.012	0.000	0.05	0.22	81.82	6694.215	20.408	4.518

Conditions: 78 nM MPO in 50 mM acetate buffer with 100 mEq/L Cl^−^ and 51 µM luminol at pH 5.0. Results of two experiments combined.

## Data Availability

Data is contained within the article.

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
