# Peer review of "Haloperoxidase-Catalyzed Luminol Luminescence"

_antioxidants, 2022, doi:10.3390/antiox11030518_

Round 1

Reviewer 1 Report

The manuscript “Haloperoxidase Catalyzed Luminol Luminescence” submitted by Robert C. Allen for publication in antioxidants describes a detailed and well elaborated study on enzymatic catalysis by haloperoxidases, in comparison to HRP, the most utilized peroxidase. The emission intensity of luminol chemiluminescence is used to measure the catalytic reaction rate where luminol is utilized as a model for oxidizable biological substrates. If I understood the information well, the kinetic data outlined here have been obtained before in studies related to a patent application (Patent US005556758A).

In my opinion the experimental results and the interpretation obtained from these kinetic data are of great importance and the manuscript should be published in antioxidants, I have only a few observations and suggestions which might be considered before publication.

Lines 27 -28: This first sentence of the introduction should be reformulated, as it appears to state that all light emission (in any chemical or biological system) is due to action of MPO. The kind of light emission in question should be specified like: "... chemiluminescence of polymorphonuclear leukocytes is the product ...."

Lines 35 -36: The sentence “Reaction of H2O2 with OCl- produces 1O2* [4-6].” Should be deleted as it contains the same information as the sentence before.

Lines 49 -60: In this part, more recent references on HRP and chloroperoxidades, as well as luminol chemiluminescence should be given.

Lines 103 -104, Figure 1: I wonder why you use here the term “velocity” for the luminol chemiluminescence “intensity”. Of course, this intensity is proportional to the reaction rate, but the term emission intensity is generally used in chemiluminescence studies. Additionally, the numerical values of counts per sec (or kcts/s, as you use) do not have any physical meaning and could be substituted by arbitrary units. Calibration of the luminometer could be achieved by using the luminol standard with hemin, to transform relative values in absolute Einstein/s values, however for the interpretation of the results this is of minor relevance.

Lines 107 -108, Figure 2: I think that, also in the reciprocal plot of the peroxide concentration, the units (e.g., 1/μmol) should be given. Additionally, the numerical values used for 1/[H2O2] should be related to the concentration like 1/μmol, It appears that in the graphics shown, relative values are used. This observation is also valid for all the other plots shown in the manuscript.

Furthermore, it is very important to point out also in the text accompanying Figure 2, that in the kinetic analysis, only data for H2O2 concentrations of lower than 6 micromolar are used.

Lines 342 -343: Just to mention it, there is a recent work showing application of 1O2 formation by reaction of chloramines with H2O2: Valdecir F. Ximenes, Thomaz P. Ximenes, Nelson H. Morgon and Aguinaldo R. de Souza Taurine Chloramine and Hydrogen Peroxide as a Potential Source of Singlet Oxygen for Topical Application. Photochemistry and Photobiology, 2021, 97: 963–970.

General: Apart from the specific points mentions above, I believe that the author should include more recent literature references which would significantly improve the quality of the manuscript.

Author Response

REVIEWER 1

My responses to Reviewer 1 comments are presented in bold italic type just below each comment.

RC Allen

The manuscript “Haloperoxidase Catalyzed Luminol Luminescence” submitted by Robert C. Allen for publication in antioxidants describes a detailed and well elaborated study on enzymatic catalysis by haloperoxidases, in comparison to HRP, the most utilized peroxidase. The emission intensity of luminol chemiluminescence is used to measure the catalytic reaction rate where luminol is utilized as a model for oxidizable biological substrates. If I understood the information well, the kinetic data outlined here have been obtained before in studies related to a patent application (Patent US005556758A).

RCA:  As per my cover letter to the Journal: The research data used to construct the manuscript Haloperoxidase Catalyzed Luminol Luminescence was performed almost thirty years ago in support of claims language for US Patent 005556758A issue date 1996. This work has not been communicated or published in the scientific literature. Engagement in my profession as a clinical pathologist has been my major distraction. Now partially retired, I have the time and enough residual intellectual capacity to properly communicate these research discoveries. The manuscript should be of interest to biochemically trained scientists, especially those interested in peroxidase enzymology.

In my opinion the experimental results and the interpretation obtained from these kinetic data are of great importance and the manuscript should be published in antioxidants, I have only a few observations and suggestions which might be considered before publication.

Lines 27 -28: This first sentence of the introduction should be reformulated, as it appears to state that all light emission (in any chemical or biological system) is due to action of MPO. The kind of light emission in question should be specified like: "... chemiluminescence of polymorphonuclear leukocytes is the product ...."

RCA:  Done.

Lines 35 -36: The sentence “Reaction of H2O2 with OCl- produces 1O2* [4-6].” Should be deleted as it contains the same information as the sentence before.

RCA:  Done.

Lines 49 -60: In this part, more recent references on HRP and chloroperoxidades, as well as luminol chemiluminescence should be given.

RCA:  This manuscript compares haloperoxidase action to peroxidase action regarding reaction order. I don’t know of any literature that describes MPO or EPO actions in such regard. To my knowledge, the cited papers by the Cormier group are the best presentation of peroxidase action (HRP) in the literature. I would be pleased to include any newer publications pertinent to the substance of the present manuscript if you are aware of any.

I also have similar data for chloroperoxidase and lactoperoxidase, but did not include them as they might dilute the extreme reaction order differences separating peroxidase action from haloperoxidase action.

Lines 103 -104, Figure 1: I wonder why you use here the term “velocity” for the luminol chemiluminescence “intensity”. Of course, this intensity is proportional to the reaction rate, but the term emission intensity is generally used in chemiluminescence studies. Additionally, the numerical values of counts per sec (or kcts/s, as you use) do not have any physical meaning and could be substituted by arbitrary units. Calibration of the luminometer could be achieved by using the luminol standard with hemin, to transform relative values in absolute Einstein/s values, however for the interpretation of the results this is of minor relevance.

RCA:  My choice of terminology probably reflects my early training in chemical kinetics and enzymology. You are correct in stating that chemiluminescence measurements are intensity measurements. Luminescence intensity (a first derivative value (dphoton/dt)) has the same meaning as luminescence velocity, e.g., RLU/s or cts/s. Chemical reactions are typically described in terms of velocity. For example, the velocity of a chemical reaction is proportional to the concentration of the reagents raised to certain power (velocity = k [A]n[B]m). Likewise, Michaelis-Menten descriptions of enzymatic activity are in terms of reaction velocity. For example, the Lineweaver-Burk (double reciprocal) plotting graphically depicts 1/Vmax (1/maximum velocity) as the point where the data line intersects the ordinate (y-axis).

I’ve described in the methods section that the luminometer reports measurements in relative light units (RLU/s), and that the luminescence velocity is an intensity value expressed as kilocounts per second (kcts/s) where a cts/s is equivalent to a RLU/s.

There are several problems with regards to quantifying light emission in as absolute photons. (This is a recognized problem and is discussed in Meth Enzymol 305: 626-629 (2000) Allen et al., Blood phagocyte luminescence: Gauging systemic immune activation.).

Lines 107-108, Figure 2: I think that, also in the reciprocal plot of the peroxide concentration, the units (e.g., 1/μmol) should be given. Additionally, the numerical values used for 1/[H2O2] should be related to the concentration like 1/μmol, It appears that in the graphics shown, relative values are used. This observation is also valid for all the other plots shown in the manuscript.

RCA: The values are not relative. The actual values are presented in the table below. I can provide all such values in tabular form if you or the editors desire.

H2O2

[H2O2]2

velocity (v)

SqRt velocity (v)

1/H2O2

1/[H2O2]2

1/v

1/√v

µmol

kcts/s

1/µmol

1/µmol2

1/kcts/s

1/√kcts/s

1.407

1.9796

431.29

20.77

0.71

1

0.0023

0.048

0.703

0.4942

154.96

12.45

1.42

2

0.0065

0.080

0.352

0.1239

48.56

6.97

2.84

8

0.0206

0.144

0.176

0.0310

13.15

3.63

5.68

32

0.0760

0.276

0.088

0.0077

3.45

1.86

11.36

129

0.2899

0.538

0.044

0.0019

0.87

0.93

22.73

517

1.1494

1.072

0.022

0.0005

0.19

0.44

45.45

2066

5.2632

2.294

0.011

0.0001

0.05

0.22

90.91

8264

20.0000

4.472

In the manuscript reactants are presented in molar mass units (e.g., µmol). For this set of experiments, the final reaction volume was 0.9 ml (in most other experiments, the reaction volume was 1.0 mL), and as such, the molar concentration was 1 µmol/0.9 mL = 1111 µM (micromolar) or 1.1 mM (millimolar). The units are presented in such manner to facilitate mass to concentration conversion. I’ve included such explanation in the Methods section.

Furthermore, it is very important to point out also in the text accompanying Figure 2, that in the kinetic analysis, only data for H2O2 concentrations of lower than 6 micromolar are used.

RCA:  I have attempted to focus attention to this detail in the legend of the Figure.

Lines 342 -343: Just to mention it, there is a recent work showing application of 1O2 formation by reaction of chloramines with H2O2: Valdecir F. Ximenes, Thomaz P. Ximenes, Nelson H. Morgon and Aguinaldo R. de Souza Taurine Chloramine and Hydrogen Peroxide as a Potential Source of Singlet Oxygen for Topical Application. Photochemistry and Photobiology, 2021, 97: 963–970.

RCA: The Ximenes et al paper is pertinent and adds to the content of the manuscript. Thank you for the information. I’ve cited and referenced it.

General: Apart from the specific points mentions above, I believe that the author should include more recent literature references which would significantly improve the quality of the manuscript.

RCA:  I’m unaware of any recent publication that would contribute information pertinent to the present manuscript. If you know of any such work, e.g., the Ximenes et al paper, please inform me.

RCA: Thank you for your thorough review.

Reviewer 2 Report

The manuscript is written well in the results section. The data presented support the conclusions regarding the common as well as haloperoxidase activities of the enzymes tested with luminal.

However, I have three major concerns:

  1. The abstract seems to lack the significance of the work and its impact. It lists a set of experimental findings without indicating why they were carried out and their importance.
  2. The introduction does not set the scene or the objectives of the work. One does not understand the impact and the relatedness of the work to the current special issue on Heme Peroxidases in (Patho)Physiological Reactions and Disease Progression. Even in the conclusions section, the author reports the findings as a bullet point and does not clearly put the results obtained in the context of disease/physiological impact of these enzymes and the reactive oxygen species.
  3. The literature citations are mainly of papers which are very old, at least from 30 odd years ago. They need to be updated.

There are a few minor typos within the text for example on page 8, line 267: produce.

Author Response

REVIEWER 2

My responses to Reviewer 2 comments are presented in bold italic type just below each comment.

RC Allen

Comments and Suggestions for Authors

The manuscript is written well in the results section. The data presented support the conclusions regarding the common as well as haloperoxidase activities of the enzymes tested with luminal.

However, I have three major concerns:

  1. The abstract seems to lack the significance of the work and its impact. It lists a set of experimental findings without indicating why they were carried out and their importance.
  2. The introduction does not set the scene or the objectives of the work. One does not understand the impact and the relatedness of the work to the current special issue on Heme Peroxidases in (Patho)Physiological Reactions and Disease Progression. Even in the conclusions section, the author reports the findings as a bullet point and does not clearly put the results obtained in the context of disease/physiological impact of these enzymes and the reactive oxygen species.
  3. The literature citations are mainly of papers which are very old, at least from 30 odd years ago. They need to be updated.

RCA:  I’m never satisfied with my writing or with the writing of most authors. I agree with your concerns and have revised the manuscript in the hope of improving it.

Regarding your 1. Comment, I’ve reconstructed the abstract and concluded that luminol luminescent measurements in acid allow specific quantification of haloperoxidase action and can be used to quantify chloride and bromide by selective use of MPO and EPO.

Regarding your 2. Comment, I’ve refocused the Introduction and removed unnecessary information.

Regarding your 3. Comment, I’ve added five additional references. If you are aware of any recent publications pertinent to the manuscript, please inform me.

There are a few minor typos within the text for example on page 8, line 267: produce.

RCA:  Corrected.

RCA: Thank you for reviewing this manuscript.

Reviewer 3 Report

Chemiluminescence of luminol was investigated in the presence of activated heme peroxidases. In this study, main focus was directed on determination of reaction order of heme peroxidases under different conditions. Myeloperoxidase and eosinophil peroxidase (as haloperoxidases) were compared to horseradish peroxidase concerning their reaction order with H2O2, halides, and luminol.

The objective of this study is badly defined and contains a wrong statement. According to the sentence in lines 63-65, the ability of MPO and EPO to exert a halogenation or peroxidase activity depends only on the applied pH. This statement is wrong. It is well known from numerous investigations that MPO and EPO can exhibit both activities at the same pH. This is mainly determined by the composition of the medium. At the end of introduction, it should briefly be indicated what is new and will be studied here in relation to previous investigations of luminol chemiluminescence in the presence of heme peroxidases.

In all figures and figure legends, there is a confusion concerning the real reactant concentrations. They are given either as amount of substance (unit pmol, nmol, µmol etc.) or as concentration (µM, mM, etc.).

Although two possible pathways for MPO- and EPO-mediated luminol light emission were mentioned, namely the pathway involving singlet oxygen and the pathway displaying a sequential reaction of luminol with HOCl and H2O2, the main focus is directed on the formation of singlet oxygen. However, no experimental evidence is given, why the singlet oxygen pathway is preferred. Data about singlet oxygen scavengers or application of media (D2O) that prolong the lifetime of 1O2 is not given.

Any oxidation of luminol by Compound I (or Compound II) of MPO and EPO is not considered in the reaction mechanism.

The first paragraph of discussion is confusing. It remains unclear why hemoglobin is mentioned here. (Patho)physiological functions of MPO are very complex and include not only participation in microbe killing. For example, in inflamed vessels MPO can diminish the bioavailability of nitrogen monoxide (NO) by oxidizing NO (this is a peroxidase activity!). No references are given in this paragraph.

Of the 30 references included, only five articles were published after the year 2000, and two after 2010.

Reference 15 is given without bibliographic data.  

Reference 14:   horseradish

Author Response

REVIEWER 3

My responses to Reviewer 3 comments are presented in bold italic type just below each comment.

RC Allen

Chemiluminescence of luminol was investigated in the presence of activated heme peroxidases. In this study, main focus was directed on determination of reaction order of heme peroxidases under different conditions. Myeloperoxidase and eosinophil peroxidase (as haloperoxidases) were compared to horseradish peroxidase concerning their reaction order with H2O2, halides, and luminol.

The objective of this study is badly defined and contains a wrong statement. According to the sentence in lines 63-65, the ability of MPO and EPO to exert a halogenation or peroxidase activity depends only on the applied pH. This statement is wrong. It is well known from numerous investigations that MPO and EPO can exhibit both activities at the same pH. This is mainly determined by the composition of the medium. At the end of introduction, it should briefly be indicated what is new and will be studied here in relation to previous investigations of luminol chemiluminescence in the presence of heme peroxidases.

RCA:  My statement is not “wrong”. This manuscript presents data and analysis in support of all positions taken and conclusions drawn. Based on luminol luminescence catalyzed by MPO at pH 5, no haloperoxidase or peroxidase action is measured in the absence of Cl- or Br-. For EPO at pH 5, no haloperoxidase or peroxidase action is measured in the absence of Br-. Likewise, based on luminol luminescence catalyzed by HRP at pH 5, no haloperoxidase or peroxidase action is measured in the presence or absence of Cl- or Br-.  

Although common peroxidase action can be measured by dye (e.g., guaiacol) dehydrogenation under acid conditions, common peroxidase catalyzed luminol luminescence does is not measurable at pH 5.

You make the nebulous statement that “This is mainly determined by the composition of the medium”. My conclusions are based on evidence presented within this manuscript, including the explicitly defined composition of all measurement media. Where is your evidence?

In all figures and figure legends, there is a confusion concerning the real reactant concentrations. They are given either as amount of substance (unit pmol, nmol, µmol etc.) or as concentration (µM, mM, etc.).

RCA:  There was an error in one of the figures that has been corrected. The quantities of reactants are described in molar mass units, e.g., pmol, nmol or µmol. The final reaction volume was either 0.9 mL or 1.0 mL as indicated. Combining the molar mass unit with the final volume give the molar concentration, e.g., a 1.0 nmol quantity in 0.9 mL is 1.0 nmol/0.9 mL or 1111 nmol/L or 1.1 µM. The molar mass units and final reaction volume are presented to facilitate conversion from molar mass to molar concentration. This information has been included in the Methods section.

Although two possible pathways for MPO- and EPO-mediated luminol light emission were mentioned, namely the pathway involving singlet oxygen and the pathway displaying a sequential reaction of luminol with HOCl and H2O2, the main focus is directed on the formation of singlet oxygen. However, no experimental evidence is given, why the singlet oxygen pathway is preferred. Data about singlet oxygen scavengers or application of media (D2O) that prolong the lifetime of 1O2 is not given.

Any oxidation of luminol by Compound I (or Compound II) of MPO and EPO is not considered in the reaction mechanism.

RCA:  I’ve expanded the manuscript to include possible non-radical singlet multiplicity pathways to luminol luminescence, including potential involvement of chloramine intermediates.

Please appreciate that the focus of this manuscript is kinetic analysis by measured reaction order, not spectroscopy.  

The first paragraph of discussion is confusing. It remains unclear why hemoglobin is mentioned here. (Patho)physiological functions of MPO are very complex and include not only participation in microbe killing. For example, in inflamed vessels MPO can diminish the bioavailability of nitrogen monoxide (NO) by oxidizing NO (this is a peroxidase activity!). No references are given in this paragraph.

RCA:  This manuscript examines the orders of the participant reactants for the haloperoxidase actions of MPO and EPO and compares them to those of common peroxidase action. You may have missed the point, but the experimental evidence of the manuscript clearly differentiates haloperoxidase and peroxidase actions based on reaction orders.

The data presented in the manuscript also show that MPO and EPO can drive luminol luminescence by common peroxidase action if alkalinity is adequate.

MPO and EPO can have multiple physiologic functions. For example, MPO, and to a lesser extent EPO, inhibit lipopolysaccharide and lipid A endotoxin activity, and such inhibition is non-enzymatic.

Of the 30 references included, only five articles were published after the year 2000, and two after 2010.

RCA:  I’ve included five additional references. If you know of any reference containing information pertinent to this manuscript, please provide them.

Reference 15 is given without bibliographic data.  

RCA:  Corrected.

Reference 14:   horseradish

RCA:  Corrected.

RCA:  Based on your statements, the content of my manuscript appear to conflict with your opinions. You have clearly expended a considerable effort reviewing it, and as such, I sincerely thank you for your effort.

Round 2

Reviewer 2 Report

I have no further comments but a general observation that it is not the job of the reviewers to up date the reference list for the author. In general, the author(s) should take care of the fact that all recent relevant literature is cited.

Author Response

Review Report Form

Open Review

English language and style

( ) Extensive editing of English language and style required
( ) Moderate English changes required
(x) English language and style are fine/minor spell check required
( ) I don't feel qualified to judge about the English language and style

Yes

Can be improved

Must be improved

Not applicable

Does the introduction provide sufficient background and include all relevant references?

( )

(x)

( )

( )

Is the research design appropriate?

(x)

( )

( )

( )

Are the methods adequately described?

(x)

( )

( )

( )

Are the results clearly presented?

(x)

( )

( )

( )

Are the conclusions supported by the results?

( )

(x)

( )

( )

Comments and Suggestions for Authors

I have no further comments but a general observation that it is not the job of the reviewers to up date the reference list for the author. In general, the author(s) should take care of the fact that all recent relevant literature is cited.

RCA:  I have again reviewed the literature and included a few additional papers and a review to the references.  

Thank you for reviewing this manuscript.

Bob Allen

28 Feb 2022

Submission Date

27 January 2022

Date of this review

25 Feb 2022 11:17:10

Reviewer 3 Report

Although the revised manuscript has been improved in some items, I’m not satisfied by all answers and corrections of the author.

The objective of this study is now better defined. My criticism concerning the original version of your manuscript was directed against your statement “Depending on the pH of the milieu, MPO and eosinophil peroxidase (EPO) can catalyze haloperoxidase action or common peroxidase action.” This statement is not true, as both peroxidases are known to catalyze both halogenation and peroxidase reactions at the same pH. Whether the catalysis of halogenation or peroxidase action predominates depends largely on composition and concentration of halides and oxidants in the reaction medium. In biochemical literature, there are numerous investigations about these issues including the determination of second order rate constants of Compound I of MPO or EPO with halides and oxidants at a given pH. Please consider, the statement “wrong” was not directed against your results.

The indication of real reactant concentrations remains confusing, as both concentration and amount of substance are inappropriately mixed. For example, abscissa of Fig. 1 is marked as [H2O2], µmol. This does not fit as the indicated unit is not a concentration unit. Or in line 119 you wrote “… responses to H2O2 concentrations below 6.0 μmol.” The same thing. Please check carefully, all figures, figure legends, and the text to avoid any inappropriate designations of molar concentrations.

There are some spelling errors and missing data.

Line 21:                 acidic

Line 61:                contact instead of contract

Line 136:              assessment

Line 276:              substrate

Line 290:              acidic

Reference 33:   incomplete bibliographic data

Author Response

Although the revised manuscript has been improved in some items, I’m not satisfied by all answers and corrections of the author.

The objective of this study is now better defined. My criticism concerning the original version of your manuscript was directed against your statement “Depending on the pH of the milieu, MPO and eosinophil peroxidase (EPO) can catalyze haloperoxidase action or common peroxidase action.” This statement is not true, as both peroxidases are known to catalyze both halogenation and peroxidase reactions at the same pH.

RCA:  As clearly stated in the revised Abstract, “Haloperoxidase catalyzed luminol luminescence requires acidity, but HRP action requires alkalinity.” This statement is correct. Luminol luminescence by peroxidase action requires alkalinity, e.g., pH 8, but haloperoxidase catalyzed luminol luminescence requires acidity, e.g., pH 5.

I did not state, nor do I believe that peroxidase catalyzed dye dehydrogenation requires alkalinity. The guaiacol assay proves this point. As presented in the Results, luminol luminescence by peroxidase action requires an alkaline pH and is not measured at pH 5. Peroxidase catalyzed guaiacol dehydrogenation within a pH range from 4 to 8.  To avoid any opportunity for misunderstanding, I’ve emphasized this point in the manuscript and have added a reference showing the broad optimal pH range for HRP catalyzed guaiacol dehydrogenation (see AL-Sa'ady et al. Int. J. Curr. Microbiol. App. Sci. 2018, 7, 328-339, doi:https://doi.org/10.20546/ijcmas.2018.706.037.)

Whether the catalysis of halogenation or peroxidase action predominates depends largely on composition and concentration of halides and oxidants in the reaction medium. In biochemical literature, there are numerous investigations about these issues including the determination of second order rate constants of Compound I of MPO or EPO with halides and oxidants at a given pH. Please consider, the statement “wrong” was not directed against your results.

RCA:  The haloperoxidase catalyzed reaction responsible for halide oxidation to hypohalite is first order for H2O2 and first order for Cl-, i.e., rate = k[H2O2]1[Cl-]1, and as such, the haloperoxidase reaction is second order overall.

The non-enzymatic reaction of H2O2 with OCl- is first order for H2O2 and first order for OCl-, i.e., rate = k[H2O2]1[OCl-]1.

The overall combined haloperoxidase mediated reactions responsible for luminol dioxygenation resulting in luminescence are second order for H2O2 and first order for luminol, i.e., rate = k[H2O2]2[luminol]1. These orders of reaction are specific for haloperoxidase catalyzed luminol luminescence that require acidity. These orders of reaction are different from the peroxidase catalyzed luminol luminescence that requires alkalinity.

The indication of real reactant concentrations remains confusing, as both concentration and amount of substance are inappropriately mixed. For example, abscissa of Fig. 1 is marked as [H2O2], µmol. This does not fit as the indicated unit is not a concentration unit. Or in line 119 you wrote “… responses to H2O2 concentrations below 6.0 μmol.” The same thing. Please check carefully, all figures, figure legends, and the text to avoid any inappropriate designations of molar concentrations.

RCA:  To remove any possible confusion, I’ve changed all the figures to show molar concentration units only. 

There are some spelling errors and missing data.

Line 21:                 acidic

(Done)

Line 61:                contact instead of contract

(Done)

Line 136:              assessment

(Done)

Line 276:              substrate

(Done)

Line 290:              acidic

(Done)

Reference 33:   incomplete bibliographic data

(Done)

Thank you for your efforts in reviewing this manuscript. I believe addressing your comments has seriously improved the manuscript.

Bob Allen

18 Feb 2022

Round 3

Reviewer 3 Report

The present version has been improved. There are some minor problems that can be addressed during proof correction.

line 179:   milliEquivalents

line 341:   occurs[RA1] ???

reference 31:  missing page numbering